# Magnitude and determinants of appropriate complementary feeding practice among mothers of children age 6–23 months in Western Ethiopia

Marga Fanta, Hirut Assaye Cherie◉*

Department of Applied Human Nutrition, Bahir Dar Institute of Technology, Bahir Dar, Ethiopia

* hirutas2000@gmail.com

## Abstract

### Background

Appropriate complementary feeding has the potential to prevent 6 percent of all under five deaths particularly in developing countries. However, infant and young child feeding practices in Ethiopia are suboptimal. Data on complementary feeding practices in Horro district are also lacking. Therefore, this study aimed to determine magnitude and determinants of appropriate complementary feeding practice among mothers of children age 6–23 months in Horro district, Western Ethiopia.

### Methods

Community based cross sectional study was conducted from February to March 2016 in six kebeles of Horro district, Western Ethiopia. A total of 325 mothers of children age 6–23 months were included in the study. Data were collected using pre-tested interviewer administered structured questionnaire; entered into EPI-INFO version 3.5.4 and analyzed using SPSS version 20. Odds ratio was calculated with 95% CI to identify determinants of appropriate complementary feeding practice. P-values less than 0.05 were considered as statistically significant.

### Results

The magnitude of appropriate complementary feeding practice in the study area was 9.91%. Lower age of child (6-11months) [AOR: 0.195, 95% CI: (0.045–0.846)], having no formal education [AOR = 0.115:95% CI: (0.002, 0.290)] and giving birth for the first time [AOR = 0.271:95% CI: (0.011, 0.463)] were factors negatively associated with appropriate complementary feeding practice.

### Conclusion

Only one tenth of mothers practiced appropriate complementary feeding. This strongly calls for sustained nutrition education targeting uneducated mothers, mothers who gave birth for the first time and those with very young children to improve the practice.

**Data Availability Statement:** All relevant data are within the paper and its Supporting Information files.

**Funding:** This research was funded by School of Research and Graduate Studies, Bahir Dar Institute of Technology through its program of funding researches conducted by its staff.

**Competing interests:** The authors have no competing interests.

**Abbreviations:** ANC, Antenatal care; AOR, Adjusted odds ratio; CI, Confidence interval; COR, Crude odds ratio; EDHS, Ethiopian Demographic and Health Survey; IYCF, Infant and Young Child Feeding; PNC, Postnatal care; SD, Standard deviation.

# Background

During the first six months of a child's life, breastfeeding is adequate to support its nutritional needs. However, after six months, the child should start to be fed nutritious and safe complementary foods [1–3]. World Health Organization (WHO) recommends exclusive breastfeeding till 6 months of age, introduction of complementary feeding at 6 month while continuing breast-feeding up to 2 years or beyond [1]. Complementary feeding is a process of starting additional food when breast milk alone is insufficient to meet the nutritional requirements of the child and therefore, other foods and liquids are needed along with breast milk or breast milk substitute [1]. The first two years of a child's life is the most important window of opportunity to prevent undernutrition and its long term adverse consequences such as impaired cognitive development, poor achievement in education and low economic productivity [2].

WHO [3] has developed four indicators to guide complementary feeding of young children. The indicators include: (1) timely introduction of complementary feeding; (2) minimum dietary diversity(feeding the child at least four or more varieties from the seven standard food groups for children aged 6–23 months); (3) minimum meal frequency (feeding the child a minimum of two or three meals with one to two snacks and three or four meals with one to two snacks per day, respectively for breastfed infants of age 6–8 and 9–23 months, and provision of milk products at least twice a day for non-breastfed infants of age 6–23 months); and (4) minimum acceptable diet (a composite indicator of minimum dietary diversity and minimum meal frequency.

The government of Ethiopia designed different strategies and programs to improve infant and young child feeding practices [4–7].The strategies and programs align with WHO's recommendations as mothers or caregivers are advised to initiate complementary feeding when the child reaches 6 months and children to be fed energy-rich, nutrient-rich, locally available and affordable foods such as thick cereals with added oil or milk, fruits, vegetables, pulses, meat, eggs, fish, and milk products.

Although breastfeeding is a common practice among the majority of Ethiopian mothers, the practice of complementary feeding is very poor [7]. For instance, the 2011 Ethiopian Demographic Health Survey indicated that about 49% of mothers initiated complementary feeding before 6 months and only 5.2% children 6–23 months old were given foods from four or more groups [8]. This same study [8] also mentioned that 27% of mothers provided various types of foods including water and butter before 6 months. Introduction of liquid and solid foods before six months increases the risk of diarrheal diseases and is an important cause of infant and young children morbidity and mortality in developing countries [1, 9]. Appropriate complementary feeding has the potential to prevent 6 percent of all under 5 deaths in developing countries [1].

Studies [10–12] from different regions of Ethiopia on appropriate complementary feeding practice reported various magnitudes and determinants. These determinants include child's characteristics (such as child's age, sex, and birth order), maternal characteristics (such as education level and postnatal care follow up) and family size. However, there is limited evidence in Horro District, Western Ethiopia with respect to complementary feeding practice and its determinants given existing regional differences in culture and food practices in Ethiopia. Therefore, in this study we have collected a cross-sectional data and determined the magnitude of appropriate complementary feeding practice and its determinants in Horro District, Western Ethiopia. The finding will help nutrition programmers to design appropriate intervention to improve complementary feeding practice in the study area.

## Methods

### Study design, setting and study population

A community based cross-sectional quantitative study was employed from February to March 2016 in Horro district. Horro district is located in Oromia Regional State, Western Ethiopia and found 315 km away from Addis Ababa, the capital city of Ethiopia. The district is one of the ten districts of Horro Guduru Wollega Zone, having both rural and urban population. The total population of the district in the year 2011/2012 was 93,129. A total of 15,301 children under five years were living in the district, of which 2,999 were children under one year. Agriculture is the main livelihood of the majority of the people while some are involved in trading. Mother-child pairs from six kebeles (five rural and one urban) were included in this study.

### Sample size and sampling procedure

Sample size was determined by using single population proportion formula with the following assumptions; 10.75% prevalence of appropriate complementary feeding from a previous study [10], 95% confidence interval, 5% margin of error, and design effect of 2. A sample size of 325 was taken after considering 10% non response rate. Stratified sampling technique was employed to divide the study area into rural and urban kebeles. In the first stage, from the twenty-three kebeles, six (5 rural and 1 urban) were selected randomly using lottery method. Then, the sample size was distributed proportionally to each kebele and individual households in the selected kebeles were selected using systematic random sampling technique after identifying an initial starting household. In the case of two eligible participants in the household, lottery method was used to select one child.

### Variables of the study

The main outcome variable of this study was appropriate complementary feeding practice. The independent variables were socio-demographic and economic characteristics and health care related characteristics such as history of antenatal care visits (ANC), place of delivery, post natal care (PNC), counseling/ advice on complementary feeding.

### Data collection procedures

Data were collected using pre-tested and interviewer administered questionnaire adapted from different literatures. The questionnaire consisted of information on background characteristics of mothers and children, maternal health care practice and child feeding practices. The questionnaire was initially prepared in English and then translated into the local language of the community, Afaan Oromo and then translated back to English to check its consistency.

Four diploma nurses were hired for data collection and two BSc nurses supervised the data collection process. Both data collectors and supervisors had prior experience on data collection and supervision and have been given further training for two days on the study objectives, purpose and interviewing techniques. Pretest was done on 32 (10%) mother- child pairs who have similar background with study participants in Amuru woreda (which is found 15 kms away from the study area) to assess the content and approach of the questionnaire. After the pretest, questions which were cited as inconsistent, invalid and poorly constructed or were likely to give unreliable data were reviewed and corrected.

**Complementary feeding indicators.** Complementary feeding practice was assessed using the key indicators recommended by WHO [3], which include timely introduction of solid, semi-solid or soft foods, minimum dietary diversity, minimum meal frequency and minimum acceptable diet calculated for the age ranges 6–11, 12–17 and 18–23 months of age based on a

24-h recall of the child's dietary intake. If a child fulfilled the above four criteria, the child was classified as having received appropriate complementary feeding.

## Data analysis

Data were cleaned, coded, entered into EPI-INFO version 3.5.4 software and transferred and analyzed using SPSS version 20. The association between single explanatory variable and dependent variable was examined through bivariate analysis, by computing odds ratio (OR) at 95% confidence level.

Variables with p-value less than 0.2 were included in the multiple logistic regression analysis to identify factors associated with appropriate complementary feeding. For all statistical significance tests between each independent and dependent variables, significance level was fixed at $P$-value $< 0.05$.

## Ethics

The study protocol was approved by the Ethical Committee of Faculty of Chemical and Food Engineering, Bahir Dar Institute of Technology (Ref.no./BiT/SCFE/ 477/2016). Informed consent was obtained verbally after explaining the study objectives for the participants. Participation was voluntary and mothers/caregivers signed (or provided a thumb print if illiterate) a statement of an informed consent after which they were interviewed.

## Results

### Socio-demographic and economic characteristics

A total of 323 respondents participated in the study with a response rate of 99.4%. The mean age of children was 15.04 months ± 4.36 (SD), and the mean age of mothers was 28.90 years with± 6.575(SD). Majority of mothers (67.5%) were farmers and about 40.9% of mothers have attended primary school. Most of the mothers (40.9%) were protestant by religion and all of them were Oromo (100%) by ethnicity (Table 1).

### Health care related characteristics

Majority of study participants (84.5%) had antenatal care follow up at least once during their last pregnancy. About 53.6% of mothers gave birth to their current child at health institution and 42.1% of mothers had received postnatal care at least one time. Among study participants, 80.5% had received advice on complementary feeding (Table 2).

### Complementary feeding practice

Table 3 shows complementary feeding practice of the respondents. Among the participants, 21.4% started complementary feeding before 6 months; 60.4% of them started at 6months and 18.2% started later after 6 months. Of the 323 participants enrolled in the study, 95.4% had ever practiced breastfeeding. The overall prevalence of children who had received minimum dietary diversity was 15.8%. Almost half of the children (51.4%) had received the minimum meal frequency and 10.5% of them received the minimum acceptable diet. The overall prevalence of appropriate complementary feeding practice in the study area was 9.91%.

Table 4 presents the seven food groups recommended by WHO for children 6-23months old. In this study, grains, roots & tubers were the most common food items consumed by children (97.8%) followed by legumes & nuts (64.7%). On the other hand, flesh foods (15.2%) and vitamin A—rich fruits & vegetables (13.9%) were foods less offered to children in the study area. The proportion of food items consumed by children increased with age of the child;

**Table 1. Socio-demographic and economic characteristics of respondents, Horro district, Western Ethiopia, 2016.**

| Variables | Frequency | Percent |
|---|---|---|
| Residence | | |
| Rural | 275 | 85.1 |
| Urban | 48 | 14.9 |
| Sex of infant/ child | | |
| Male | 148 | 45.8 |
| Female | 175 | 54.2 |
| Mother's age | | |
| $\leq$ 20 | 48 | 14.9 |
| 21–25 | 56 | 17.3 |
| 26–30 | 93 | 28.8 |
| $\geq$ 31 | 126 | 39 |
| Child's age(months) | | |
| 6–11 | 72 | 22.3 |
| 12–17 | 148 | 45.8 |
| 18–23 | 103 | 31.9 |
| Sex of child | | |
| Male | 148 | 45.8 |
| Female | 175 | 54.2 |
| Family size | | |
| 2–3 | 41 | 12.7 |
| 4–5 | 117 | 36.2 |
| $\geq$ 6 | 165 | 51.1 |
| Birth order | | |
| First | 59 | 18.3 |
| Second | 60 | 18.6 |
| Third and above | 204 | 63.1 |
| Marital status | | |
| Married | 298 | 92.3 |
| Divorced | 12 | 3.7 |
| Others | 13 | 4 |
| Religion | | |
| Protestant | 158 | 48.9 |
| Orthodox | 93 | 28.8 |
| Wakefata | 46 | 14.2 |
| Muslim | 26 | 8.1 |
| Educational status of mother | | |
| No formal education | 111 | 34.3 |
| Primary | 132 | 40.9 |
| Secondary | 48 | 14.9 |
| Above secondary | 32 | 9.9 |
| Educational status of father | | |
| No formal education | 78 | 24.2 |
| Primary school | 154 | 47.7 |
| Secondary | 46 | 14.2 |
| Diploma and above | 45 | 13.9 |
| Occupational status of mother | | |
| Farmer | 219 | 67.8 |

(*Continued*)

**Table 1.** (Continued)

| Variables | Frequency | Percent |
|---|---|---|
| Merchant | 27 | 8.4 |
| Employee | 22 | 6.8 |
| House wife | 55 | 17 |
| Household monthly income | | |
| (Ethiopian Birr) | | |
| $\leq$ 999 | 187 | 57.9 |
| 1000–1999 | 72 | 22.3 |
| 2000–2999 | 29 | 9 |
| $\geq$ 3000 | 35 | 10.8 |

children in the age group of 6–11 months were offered less variety of foods compared to children in the age group of 18–23 months.

## Determinants of appropriate complementary feeding practice

Both bivariate and multivariate analyses were performed to identify determinants of appropriate complementary feeding practice (Table 5). The bivariable logistic regression analysis revealed the presence of association between appropriate complementary feeding practice and residence, age of child, place of birth, educational status of mother, educational status of father, occupational status of mother, PNC follow up and child's birth order.

In multiple logistic regression analysis, only three variables were significantly associated with appropriate complementary feeding; child's age, mother's educational status and child's birth order. Children with age category of 6-11months were 80.5% times less likely to be fed appropriately than those with age category of 18–23 months [AOR = 0.195;95% CI:(0.045, 0.846)]. Those mothers who had no formal education were 88.5% times less likely to practice appropriate complementary feeding than those who had educational status above secondary school [AOR = 0.115;95% CI: (0.002,0.290)]. Besides, mothers who gave birth for the first time were 72.9% times less likely to practice appropriate complementary feeding than those mothers who gave birth for the third time and above [AOR = 0.271:95% CI:(0.011,0.463)].

**Table 2. Health care related characteristics of respondents, Horro district, Western Ethiopia, 2016.**

| Variables | Frequency | Percent |
|---|---|---|
| Place of delivery | | |
| Home | 150 | 46.4 |
| Health institute | 173 | 53.6 |
| ANC follow up | | |
| Yes | 273 | 84.5 |
| No | 50 | 15.5 |
| PNC follow up | | |
| Yes | 136 | 42.1 |
| No | 187 | 57.9 |
| Advised/counseled on complementary feeding | | |
| Yes | 260 | 80.5 |
| No | 63 | 19.5 |

**Table 3. Complementary feeding practice of mothers, Horro district, Western Ethiopia, 2016.**

| Variables | Category | Frequency | Percent |
|---|---|---|---|
| Age of starting complementary feeding | Before 6 months | 69 | 21.4 |
|  | At 6 months | 195 | 60.4 |
|  | Later after 6 months | 59 | 18.2 |
| Currently breastfeeding | Yes | 163 | 50.5 |
|  | No | 160 | 49.5 |
| Timely introduction of complementary feeding | Yes | 195 | 60.4 |
|  | No | 128 | 39.6 |
| Minimum dietary diversity | Yes | 51 | 15.8 |
|  | No | 272 | 84.2 |
| Minimum meal frequency | Yes | 166 | 51.4 |
|  | No | 157 | 48.6 |
| Minimum acceptable diet | Yes | 34 | 10.5 |
|  | No | 289 | 89.5 |
| Appropriate complementary feeding | Yes | 32 | 9.91 |
|  | No | 291 | 90.09 |

## Discussion

The magnitude of appropriate complementary feeding practice in the study area was 9.91%. This is almost similar with studies reported from Northern and Southern Ethiopia [10, 11]. However, the result was lower than a study reported from Northern Ghana [13]. Our finding shows that most of the mothers did not practice appropriate complementary feeding although the majority of mothers responded that they have got counseling on complementary feeding. Effective social and behavior change communication should be therefore designed and implemented to improve the situation.

In this study, only 60.4% of mothers had initiated complementary feeding at the 6th month of the child's age as recommended by WHO [3]. This result was consistent with findings reported from Eastern Ethiopia [12] but higher than the national prevalence [8]. On the other hand, the result was lower than studies reported from Northern (79.7%) [10] and Southern (72.5%) Ethiopia [11]. About 40% of mothers in our study started complementary feeding either early before 6 months or later at seven month or beyond. Studies indicated that both early and late initiation of complementary feeding could result in deterioration in physical growth of infants and children and predisposes them to various infectious diseases which further exacerbate their nutritional status [10, 11]. Our finding thus calls for the need to address the problem effectively in the region to reduce the risk of growth faltering among infants and children in the study area.

**Table 4. Types of foods given to children age 6–23 months, Horro district, Western Ethiopia, 2016.**

| Food groups | Food items given to children | | | |
|---|---|---|---|---|
|  | 6-11months (n = 72) | 12-17months (n = 148) | 18–23 months (n = 103) | 6-23months (n = 323) |
| Grains, roots and tubers | 69(95.8) | 145(98) | 102(99) | 316(97.8) |
| Legumes and nuts | 39(54.2) | 102(68.9) | 68(66) | 209(64.7) |
| Dairy products | 29(40.3) | 66(44.6) | 56(54.4) | 151(46.7) |
| Flesh foods | 8(11.1) | 20(13.5) | 21(20.6) | 49(15.2) |
| Eggs | 25(34.7) | 59(39.9) | 45(43.7) | 129(39.9) |
| Vitamin A rich fruits and vegetables | 6(8.3) | 17(11.5) | 22(21.4) | 45(13.9) |
| Other fruits and vegetables | 23(31.9) | 45(30.4) | 36(35.3) | 104(32.3) |

**Table 5. Bivariate and multivariate logistic regression analysis.**

| Variables | Complementary feeding practice | | Crude OR(95% CI) | Adjusted OR (95% CI) |
|---|---|---|---|---|
| | Appropriate | Inappropriate | | |
| | N (%) | N (%) | | |
| Resident | | | | |
| Rural | 27(9.8) | 248(90.2) | 1 | 1 |
| Urban | 10(20.8) | 38(79.2) | 0.414(0.186,0.922) | 0.664(0.198,2.233) |
| Age of child* | | | | |
| 6–11 | 3(4.2) | 69(95.8) | 1 | 1 |
| 12–17 | 17(11.5) | 131(88.5) | 0.656(0.318,1.356) | 0.502(0.194,1.300) |
| 18–23 | 17(16.5) | 86(83.5) | 0.220(0.062,0.781) | 0.195(0.045,0.846) |
| Place of birth | | | | |
| Home | 7(4.7) | 143(95.3) | 1 | 1 |
| Health institution | 30(17.3) | 143(82.7) | 0.233(0.099,0.549) | 0.353(0.109,1.139) |
| Occupational status of mother | | | | |
| Farmers | 19(8.7) | 200(91.3) | 1 | 1 |
| Merchant | 3(11.1) | 24(88.9) | 0.500(0.127,1.968) | 0.199(0.032,1.231) |
| Employed | 4(18.2) | 18(81.8) | 0.889(0.250,3.162) | 0.223(0.039,1.293) |
| House wives | 11(20.0) | 44(80.0) | 0.380(0.169,0.855) | 0.314(0.098,1.005) |
| Educational status of mother* | | | | |
| No formal education | 2(1.8) | 109(98.2) | 1 | 1 |
| Primary | 13(9.8) | 119(90.2) | 0.568(0.210,1.531) | 0.193(0.049,0.765) |
| Secondary | 11(22.9) | 37(77.1) | 0.209(0.083,0.527) | 0.190(0.027,0.364) |
| Above secondary | 11(34.4) | 21(65.6) | 0.035(0.007,0.170) | 0.115(0.002,0.290) |
| Educational status of father | | | | |
| No formal education | 7(9.0) | 71(91.0) | 1 | 1 |
| Primary | 15(9.7) | 139(90.3) | 1.095(0.427,2.807) | 0.498(0.130,1.897) |
| Secondary | 5(10.9) | 41(89.1) | 1.237(0.369,4.149) | 0.301(0.056,1.616) |
| Above secondary | 10(22.2) | 35(77.8) | 2.898(1.017,8.259) | 0.639(0.148,2.764) |
| Postnatal care | | | | |
| Yes | 26(19.1) | 110(80.9) | 3.782(1.797,7.959) | 2.253(0.875,5.799) |
| No | 11(5.9) | 176(94.1) | 1 | 1 |
| Birth order* | | | | |
| First | 2(3.4) | 57(96.6) | 0.204(0.047,0.878) | 0.271(0.011,0.463) |
| Second | 5(8.3) | 55(91.7) | 0.527(0.195,1.425) | 0.333(0.090,1.243) |
| ≥ third | 30(14.7) | 174(85.3) | 1 | 1 |

* p<0.05.

About 51.5% of children had the minimum meal frequency and this is similar with a study done in two rural zones of Ethiopia [14]. However, this magnitude was lower than those reported from Southeast Asian countries such as Sri Lanka [15], Bangladesh [16] and Nepal [17]. This variation might be due to the low economic status of mothers as most of the mothers included in this study had monthly income <40 USD. Besides, the majority of mothers were farmers and work away from home which could be a challenge to exercise appropriate child care practices. Children should get the minimum number of meals per day to increase the probability of reaching the required levels of energy and micronutrient intakes. As such, for healthy breastfed infants, complementary foods should be provided 2–3 times per day at 6–8 months of age, 3–4 times per day at 9–11 and 12–24 months of age, with additional nutritious snacks offered 1–2 times per day as desired [18, 19].

Our study showed that only 15.8% of children in the study area have got the minimum dietary diversity. This finding is almost similar with studies reported from Northern Ethiopia [10] and India [20]. However, the result was lower than studies reported from two zones of rural Ethiopia [14] and other countries such as Sir Lanka [15], Bangladesh [16], Nepal [17], Kenya [21], Northern Ghana [13] and Tanzania [22]. This low proportion of dietary diversity in the study area might be due to socio economic factors and lack of awareness by mothers on the importance of diversifying diet in feeding children.

The proportion of children who received the minimum acceptable diet was 10.5%. Our result is similar with findings reported from Northern Ethiopia [10], two rural zones of Ethiopia [14] and Tanzania [22], but higher than the national proportion [8]. On the other hand, result was lower than findings reported from Northern Ghana [13], Sir Lanka [15], Bangladesh [16] and Nepal [17]. Our finding shows that children in the study area are not getting adequate energy and nutrients appropriate for their age.

Our study revealed that the proportion of appropriate complementary feeding practice has increased as educational status of mothers increased. This finding was in line with studies done in Northern Ethiopia [10] and Northern Ghana [13]. Mothers with no formal education were not practicing appropriate complementary feeding and this was consistent with findings reported from other countries; India [20], Nairobi [21], Tanzania [22], Indonesia [23], Sir Lanka [15], Bangladesh [16] and Nepal [17].

Children within the age group of 6-11months were less likely to be appropriately fed as compared to children in the age group of 18–23 months. This result was consistent with findings reported from Nepal [17], Northern Ghana [13], Tanzania [22] and Indonesia [23]. This might be due to the misunderstanding among mothers that young children cannot digest foods like meat, eggs as well as fruits and vegetables. Therefore, emphasis should be given on the need for improving the dietary quality of complementary foods such as inclusion of animal source foods as well as vitamin A rich vegetables and fruits in the diet of children.

Mothers who gave birth for the first time were less likely to practice appropriate complementary feeding than those who gave birth more than three times. Although lack of experience could be one of the reasons for this, the situation could have been averted if appropriate messages on child care practices were given to mothers during their antenatal as well as post natal care visits.

Our study had two major limitations. The cross sectional nature of the study prevents it to establish a cause and effect relationship. Though a number of measures were taken, there might be a possibility to recall bias some variables such as time to introduction of complementary foods and type of foods given to the child the day before data collection.

## Conclusion

This study showed that only one tenth of mothers in the region were practicing appropriate complementary feeding. This may have a negative impact on the subsequent growth and development of children. Determinants that consistently associated negatively with appropriate complementary feeding practice in the study area were age of the child specifically lower age, birth order and mother's education. Interventions on appropriate complementary feeding should therefore target uneducated mothers, mothers who gave birth for the first time and those with young children.

## Supporting information

**S1 File.**
(DOC)

**S2 File.**
(SAV)

## Acknowledgments

We are grateful to Horro District Health Office for facilitating the research process by timely writing support letters. Our special thanks also go to data collectors and study participants.

## Author Contributions

**Conceptualization:** Marga Fanta, Hirut Assaye Cherie.

**Data curation:** Marga Fanta.

**Formal analysis:** Marga Fanta, Hirut Assaye Cherie.

**Investigation:** Marga Fanta.

**Methodology:** Marga Fanta, Hirut Assaye Cherie.

**Supervision:** Hirut Assaye Cherie.

**Writing – original draft:** Marga Fanta.

**Writing – review & editing:** Hirut Assaye Cherie.

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
