## [Decision Letter · Decision Letter 0]

26 May 2020

PONE-D-20-06067

Determinants of appropriate complementary feeding practice among mothers of children age 6-23 months in Western Ethiopia

PLOS ONE

Dear Dr.  Cherie

Thank you for submitting your manuscript to PLOS ONE. After careful consideration, we feel that it has merit but does not fully meet PLOS ONE’s publication criteria as it currently stands. Therefore, we invite you to submit a revised version of the manuscript that addresses the points raised during the review process.

We look forward to receiving your revised manuscript.

Kind regards,

Choolwe Jacobs, PhD

Academic Editor

PLOS ONE

Additional Editor Comments:

The authors should ensure a thorough review of the manuscript and additionally attend to all typo and grammatical errors. The authors should also adequately demonstrate the additional value/contribution this study brings considering that many other similar studies have been done in Ethiopia.

2. Please include additional information regarding the survey or questionnaire used in the study and ensure that you have provided sufficient details that others could replicate the analyses. For instance, if you developed a questionnaire as part of this study and it is not under a copyright more restrictive than CC-BY, please include a copy, in both the original language and English, as Supporting Information. Moreover, please include more details on how the questionnaire was pre-tested, and whether it was validated.

5. Please amend your list of authors on the manuscript to ensure that each author is linked to an affiliation. Authors’ affiliations should reflect the institution where the work was done (if authors moved subsequently, you can also list the new affiliation stating “current affiliation:….” as necessary).

6. We note you have included a table to which you do not refer in the text of your manuscript. Please ensure that you refer to Table 5 in your text; if accepted, production will need this reference to link the reader to the Table.

7. Your ethics statement must appear in the Methods section of your manuscript. If your ethics statement is written in any section besides the Methods, please move it to the Methods section and delete it from any other section. Please also ensure that your ethics statement is included in your manuscript, as the ethics section of your online submission will not be published alongside your manuscript.

Reviewers' comments:

Reviewer's Responses to Questions

**Comments to the Author**

1. Is the manuscript technically sound, and do the data support the conclusions?

Reviewer #1: Partly

Reviewer #2: Yes

2. Has the statistical analysis been performed appropriately and rigorously? 

Reviewer #1: No

Reviewer #2: Yes

3. Have the authors made all data underlying the findings in their manuscript fully available?

Reviewer #1: Yes

Reviewer #2: Yes

4. Is the manuscript presented in an intelligible fashion and written in standard English?

Reviewer #1: No

Reviewer #2: Yes

5. Review Comments to the Author

Reviewer #1: GENERAL COMMENTS

Determining the complementary feeding practices is critical for children aged six months and above. Therefore, this study is very critical in addressing behaviour change regarding best practices among the population of interest. This being the case, the arguments in this study are very weak in informing policy.

The grammar is generally poor and requires an English language editor’s services.

There are WHO guidelines on breastfeeding for HIV exposed infants and the rest of the population that are clear when each feeding method can be initiated and discontinued, however, this study has not referred to these.

ABSTRACT

If a researcher chose to read only the abstract, then they would consider the paper very inadequate in addressing the gap in research.

Lines 23-24, the authors said: “The magnitude of appropriate complementary feeding practice in the study area was very low”. What would you consider low.

BACKGROUND

Much as the background needs to be brief, it has very little to reflect the gap that this research is addressing.

There are certain questions one would like to answer: What is complementary feeding practice? Is it a collection of different activities or it is one event? This is not clear in the background.

The authors will do well to give a clear explanation of the food types as they appear in the national guidelines and also based on food types in the region. Any brief explanation of these groups and maybe what do they comprise of in Ethiopia. It would help also to provide information on guidelines relating to initiation of complementary foods in Ethiopia and also speaking to WHO guidelines.

What is context specific information?

OVERALL AIM

Lines 50-51-the authors collected data to determine the magnitude of appropriate complementary feeding practice and its determinants in Western Ethiopia to guide policy actions in the area.

This aim is two pronged because the authors can come up with just results on magnitude of complementary feeding practice in the population of interest….but also they can establish determinants of complementary feeding practice.

Handling the study in this way made it difficult for the authors to make scientifically based conclusions as shown later in the comments. Either one of the issues studied separately would provide evidence to inform policy.

For this study, the main outcome measure (lines 76 & 77) do not relate with the aim of the study. Therefore, the authors were not mindful of the importance of answering the research question adequately and might benefit from recasting the overall aim to tally with what the data is describing.

METHODOLOGY

Here it would help to describe what constitutes appropriate complementary feeding and its components

Under variables of the study, the authors mention that the mothers identified for this study should have ANC visits, delivered at the health facility and attended PNC with additional information on feeding counselling. The link is unclear, but how do you link this care to magnitude of appropriate complementary feeding?

Data collection procedures: Instead of saying diploma or BSc nurses, one would think describing their skills is better than their qualification.

RESULTS

This section of the manuscript should be reorganized based on the overall aim after recasting it.

DISCUSSION

This statement should open with the main outcome of the study to see that the research question has been answered.

Lines 208-209-This statement contradicts with the WHO recommendation of exclusive breastfeeding up to six months. This is worrying.

Lines 209-201-What would you consider to be premature for your study?

This comparison might not suffice because of differences in characteristics.

Lines 228-231: sentence is too long and needs recasting to make the message clear in this context.

Our finding shows that children in the study area as well as in Ethiopia in general are not getting adequate energy and nutrients compared to children in many African and Asian countries: This sentence requires a source; I doubt the authors have authority to this information.

Lines 256-257: This type writing is all over the discussion and it has become monotonous. The authors can vary the expression.

CONCLUSION

Lines 271-273: When you compare this statement with the aim of the study and the background, the authors appear to be contradicting themselves or tying themselves. Their study does not appear to be different from the survey they referred to in the background.

The recommendation: ‘Interventions on appropriate complementary feeding should therefore target uneducated mothers, mothers who gave birth for the first time and those with young children’.

If the idea is to improve the practices among the uneducated women, redesigning and packaging the intervention to suite this population then ties in with what authors earlier indicated in line 49-50 ‘context specific information to design appropriate interventions’

Reviewer #2: The study is well written and detailed and addresses its objectives..on the other hand their are many other similar studies conducted in different parts of Ethiopia . So that the authors should describe the contribution of their study to the existing literature.

6. PLOS authors have the option to publish the peer review history of their article (what does this mean?). If published, this will include your full peer review and any attached files.

Reviewer #1: No

Reviewer #2: No

---

## [Author Response · Author response to Decision Letter 0]

29 Jun 2020

The response for editor's and reviewers' comments are included in this submission under' Response to reviewers'. Thanks

---

## [Editor Report · Decision Letter 1]

30 Jul 2020

PONE-D-20-06067R1

Magnitude and determinants of appropriate complementary feeding practice among mothers of children age 6-23 months in Western Ethiopia

PLOS ONE

Dear Dr.Hirut Assaye Cherie

Thank you for submitting your manuscript to PLOS ONE. After careful consideration, we feel that it has merit but does not fully meet PLOS ONE’s publication criteria as it currently stands. Therefore, we invite you to submit a revised version of the manuscript that addresses the points raised during the review process.

We look forward to receiving your revised manuscript.

Kind regards,

Choolwe Jacobs, PhD

Academic Editor

PLOS ONE

---

## [Author Response · Author response to Decision Letter 1]

15 Aug 2020

Here we have attached 'response to reviewers' document to address reviewers ' comments

---

## [Decision Letter · Decision Letter 2]

8 Dec 2020

Magnitude and determinants of appropriate complementary feeding practice among mothers of children age 6-23 months in Western Ethiopia

PONE-D-20-06067R2

Dear Dr. Cherie,

We’re pleased to inform you that your manuscript has been judged scientifically suitable for publication and will be formally accepted for publication once it meets all outstanding technical requirements.

Kind regards,

Emily A Hurley, M.P.H., Ph.D.

Academic Editor

PLOS ONE

Additional Editor Comments (optional):

Reviewers' comments:

Reviewer's Responses to Questions

**Comments to the Author**

1. If the authors have adequately addressed your comments raised in a previous round of review and you feel that this manuscript is now acceptable for publication, you may indicate that here to bypass the “Comments to the Author” section, enter your conflict of interest statement in the “Confidential to Editor” section, and submit your "Accept" recommendation.

Reviewer #1: All comments have been addressed

Reviewer #2: All comments have been addressed

2. Is the manuscript technically sound, and do the data support the conclusions?

Reviewer #1: Yes

Reviewer #2: Yes

3. Has the statistical analysis been performed appropriately and rigorously? 

Reviewer #1: Yes

Reviewer #2: Yes

4. Have the authors made all data underlying the findings in their manuscript fully available?

Reviewer #1: Yes

Reviewer #2: Yes

5. Is the manuscript presented in an intelligible fashion and written in standard English?

Reviewer #1: Yes

Reviewer #2: Yes

6. Review Comments to the Author

Reviewer #1: The authors endeavored to respond to the issues I raised in my first review. I am convinced that the editor will find this manuscript important for the rest of the research community and add to the body of knowledge in this area.

Reviewer #2: They tried to answer the questions raised during the first review process and the tried to improve the language.Overall, They have addressed my concerns.

7. PLOS authors have the option to publish the peer review history of their article (what does this mean?). If published, this will include your full peer review and any attached files.

Reviewer #1: No

Reviewer #2: No

---

## [Editor Report · Acceptance letter]

18 Dec 2020

PONE-D-20-06067R2 

Magnitude and determinants of appropriate complementary feeding practice among mothers of children age 6-23 months in Western Ethiopia 

Dear Dr. Cherie:

I'm pleased to inform you that your manuscript has been deemed suitable for publication in PLOS ONE. Congratulations! Your manuscript is now with our production department. 

Kind regards, 

on behalf of

Dr. Emily A Hurley 

Academic Editor

PLOS ONE